# Screening and Mechanism of Novel Angiotensin-I-Converting Enzyme Inhibitory Peptides in *X. sorbifolia* Seed Meal: A Computer-Assisted Experimental Study Method

**DOI:** 10.3390/molecules27248792

**Published:** 2022-12-12

**Authors:** Yihan Mu, Dongwei Liu, Huaping Xie, Xinyu Zhang, Xue Han, Zhaolin Lv

**Affiliations:** 1College of Biological Sciences and Biotechnology, Beijing Forestry University, Beijing 100083, China; 2Urban and Rural Development Research Center, Beijing 100835, China

**Keywords:** *X. sorbifolium*, angiotensin-I-converting enzyme, computer-assisted, LC-MS/MS, molecular docking

## Abstract

Angiotensin-I-converting enzyme (ACE) inhibitors are used extensively to control hypertension. In this study, a computer-assisted experimental approach was used to screen ACE-inhibiting peptides from *X. sorbifolum* seed meal (XSM). The process conditions for XSM hydrolysis were optimized through the orthogonal experimental method combined with a database. The optimal conditions for ACE inhibition included an alkaline protease dose of 5%, 45 °C, 15 min and pH 9.5. The hydrolysate was analyzed by LC-MS/MS, and 10 optimal peptides were screened. Molecular docking results revealed four peptides (GGLPGFDPA, IMAVLAIVL, ETYFIVR, and INPILLPK) with ACE inhibitory potential. At 0.1 mg/mL, the synthetic peptides GGLPGFDPA, ETYFIVR, and INPILLPK provided ACE inhibition rates of 24.89%, 67.02%, and 4.19%, respectively. GGLPGFDPA and ETYFIVR maintained high inhibitory activities during in vitro digestions. Therefore, the XSM protein may be a suitable material for preparing ACE inhibitory peptides, and computer-assisted experimental screening is an effective, accurate and promising method for discovering new active peptides.

## 1. Introduction

Angiotensin-I-converting enzyme (ACE; EC 3.4.15.1) is a nonspecific dipeptidyl carboxypeptidase that was originally isolated from horse blood and is related to the regulation of the renin-angiotensin system by blood pressure [1,2]. ACE inhibition is considered to be an effective therapeutic strategy to treat hypertension. Potent synthetic ACE inhibitors, such as captopril, enalapril, and lisinopril, are widely used as clinical treatments for hypertension in humans. However, synthetic ACE inhibitors may cause several undesirable side effects, such as coughing, allergic reactions, the loss of taste, renal impairment, angioneurotic edema, and skin rashes [3,4,5]. ACE inhibitors derived from food have safety advantages over synthetic compounds. Thus, natural, and nontoxic ACE inhibitors, are derived from livestock, grain, fish, etc., and these inhibitors can be used as safe and economic substitutes for synthetic and economical purposes. In recent years, many researchers have isolated ACE inhibitory peptides from a variety of foods, such as dairy products [6], fermented products [7], grains [8], and marine organisms [9]. Until now, no report has involved the extraction of protein and the preparation of ACE inhibitory peptides from the seed meal of *X. sorbifolia*.

*X. sorbifolium* is a perennial shrub that belongs to the Sapindaceae family, and it has been cultivated for a long time in Northeastern and Northern China for its oil and for traditional medicine [10]. *X. sorbifolia* can be used to treat arteriosclerosis, hyperlipemia, hyperpiesia, chronic hepatitis, rheumatism, and enuresis [11]. *X. sorbifolium* seed meal (XSM), a coproduct after oil extraction from seeds, is rich in nutrients and has a protein content of 31–37%. The amino acid composition of the XSM proteins is similar to that of soybean proteins [10], so it is speculated that the protein hydrolysis peptides from XSM may exhibit ACE inhibitory activity similar to that of hydrolysis peptides from soy protein [12]. In previous studies, the sequences of ACE inhibitory peptides from different biological sources differed greatly, which directly affected the conformational relationship with ACE. Therefore, exploring the sequence of inhibitory peptides and the conformational relationship with ACE is crucial for the application of inhibitory peptides.

The traditional methods for obtaining ACE inhibitory peptides are based on extensive experiments. The necessary hydrolytic enzymes are first obtained by experimental analysis and then purified using ultrafiltration, gel filtration chromatography, and reversed-phase high-performance liquid chromatography to obtain the target peptides [13,14]. This method is time-consuming and expensive and results in a low yield of bioactive peptides. With the development of big data and molecular docking technology, computer-assisted experimental screening of bioactive peptides has emerged. In contrast, computer-assisted screening of ACE-inhibiting peptides offers significant advantages [15]. In addition, computer assistance can be applied throughout the process of screening bioactive peptides. For example, BLAST and BIOPEP-UWM are used in the process of screening protein hydrolases for enzyme optimization [16]; ProtParam, ToxinPred, and ExPASy can be used to predict the molecular weight, isoelectric point, toxicity, and the instability index of peptides [2]; Discovery Studio and AutoDock are used for molecular docking to predict the activity of inhibitory peptides [17]. To date, many ACE inhibitory peptides have been successfully screened from natural proteins by computer-assisted methods, and these peptides include camel hemorphins [18], quinoa bran albumin [19], and Atlantic salmon proteins [20]. All these studies show that computer-assisted experiments are an efficient and inexpensive method for screening ACE inhibitory peptides in proteins.

The aim of this study was to isolate and characterize a novel peptide with high ACE inhibitory activity from XSM proteins using a computer-assisted experimental method and to investigate the mechanism of action. To investigate the optimal conditions for generating XSM hydrolysis, we used the database (BIOPEP and UniProt) to simulate and screen alkaline proteases as hydrolytic enzymes. Orthogonal experimental design (OED) was used to optimize the conditions under which hydrolysis takes place at different pH values, temperatures, amounts of protease and times on the degree of hydrolysis (DH) and ACE inhibitory activities. The prepared ACE inhibitory peptides were then evaluated. The peptide samples were analyzed using LC-MS/MS, and the amino acid sequences of the best ACE inhibitory peptides and their positions in the proteins were screened in combination with the protein database and the data processing software ProteinPilot™ Software 5.0. The toxicity and sensitization of the peptides were evaluated using Toxinpred and AllerTOP v.2. In addition, the mechanism of peptide and ACE interactions was explored by molecular docking, and the binding sites and spatial effects of the inhibitory peptides were discussed. Finally, we synthesized peptides in vitro based on the molecular docking results and verified their ACE inhibitory activity and biostability under in vitro digestion models. This study provides a new idea for computer-assisted experimental screening of bioactive peptides, and using this method, novel peptides with ACE inhibitory potential were successfully screened from the seed meal proteins of XSM.

## 2. Results and Discussion

### 2.1. Enzyme Selection

As shown in Table 1, alkaline protease had the highest hydrolysis for several albumins selected at UniProt Knowledgebase at 25.58%, 16.19%, and 24.35%, respectively, which means that alkaline protease has a wider range of active sites for these three proteins. Among the five proteases, alkaline protease had the highest AE for 2SS_SOYBN and ALB1_PSOTE with 0.0227 and 0.0155, respectively, and Plasmin had the highest AE for ALB1_SOYBN with 0.0156. Notably, alkaline protease possessed a similar AE to Plasmin for ALB1_SOYB with 0.0155. Based on the above results, alkaline protease was selected to hydrolyze the XSM protein in order to obtain more active and more bioactive peptides.

### 2.2. Determination of the Optimal Hydrolysis Process

#### 2.2.1. Single-Factor Experiments

According to previous studies, extraction temperature and time, enzyme dosage, and pH, are important variables affecting protein DH [21,22]. To obtain as many hydrolyzed peptides as possible, we evaluated the effect of the combination of these variables on DH. As shown in Figure 1A, when the temperature was lower than 40 °C, the enzyme activity was enhanced with increasing temperature; however, when the temperature was higher than 40 °C, the DH decreased gradually due to the inhibition of enzyme activity [23]. The DH of the protein was the highest at 40 °C; thus, 35 °C, 40 °C, and 45 °C, were selected for the subsequent orthogonal test.

Figure 1B shows that the DH of the protein rapidly rose to 12.34% and reached equilibrium. The reason is that there are many binding sites between the enzyme and the substrate at the beginning of the reaction, so the enzymatic products can be generated quickly [24]. When the time was longer than 20 min, the degree of hydrolysis slowly decreases, which is due to the pH of the reaction system gradually decreasing as the enzymatic digestion proceeds, and when the pH decreases below 9.0, the activity of alkaline protease is inhibited [25]. From the perspective of saving time and experimental cost, 15 min, 20 min, and 25 min were selected for the next experiment.

Figure 1C indicates that the DH of the protein increased until the enzyme addition reached 4%, and the growth rate of DH became slow. The reason is that when too much enzyme is added, there is not enough substrate to combine with the enzyme [26]. Considering the cost savings of the enzyme, 3%, 4%, and 5%, were selected for the orthogonal test.

The effect of pH on DH is illustrated in Figure 1D. When the pH of the solution increased from 8.0 to 9.0, the DH of the protein rapidly increased. When the pH was 9.0, the DH of the protein reached a maximum. Proteolysis should be carried out under conditions where enzymatic activity is optimal, but as pH rises beyond the optimal range of enzymatic activity, enzymatic activity is inhibited, thereby hindering the production of hydrolysates [23,25]. Therefore, the pH values of 8.5, 9.0, and 9.5 were selected for the orthogonal test.

#### 2.2.2. Orthogonal Experiments

According to the results of the single-factor experiment, orthogonal experiments were carried out to obtain the optimal enzymatic hydrolysis conditions, and the experimental results are shown in Appendix A. The best process condition was A_3_B_2_C_2_D_1_, that is, the enzyme dosage was 4%, the temperature was 45 °C, the time was 25 min, and the pH was 8.5. Under these conditions, the DH of the protein was the largest at 12.83%.

However, the results obtained by using only the index of hydrolysis as the optimization condition are one-sided. As an evaluation index in single-factor experiments, DH was used to obtain more kinds of peptides, but ultimately, the aim was to obtain peptides with ACE inhibition rates. Therefore, we determined the ACE inhibition rates of the nine groups of protein hydrolysates in Appendix A. We found that the best ACE inhibition rate process conditions were A_1_B_3_C_3_D_3_, that is, the enzyme dosage was 5%, the temperature was 45 °C, the time was 15 min, and the pH was 9.5. Under these conditions, the hydrolysate ACE inhibition rate was 99.28%. These results indicate that the DH of the XSM protein is not related to the ACE inhibitory activity of the hydrolysates, which is consistent with the findings of Tian et al. [27]. To obtain more peptides with ACE inhibitory activity, we chose A_1_B_3_C_3_D_3_ as the final hydrolysis conditions for the hydrolysates of the XSM protein.

### 2.3. LC-MS/MS Screening to Identify the Structure of the ACE Inhibitory Peptide

The secondary mass spectra (MS/MS) of the parent ions were resolved, and the backbone fragment ions were classified according to their different break sites in the peptide, including sequence ions, intermediate fragment ions, and satellite ions. The positive charge is retained at the N-terminal end, and the a, b, and c series ions are obtained; if the positive charge is retained at the C-terminal end of the fragment ion, the x, y, and z series ions are obtained, while the b and y series ion fragments are commonly used in peptide sequence analysis.

The original mass spectrometry database was compared with the UniProt-Xanthoceras sorbifolia database using the ProteinPilot™ Software 5.0 software. The peptides were selected according to the following screening principles: (1) first, it was determined that the false discovery rate (FDR) for the false positive rate was below 1% and the peptides before the appearance of the first RRRRR were preferred; (2) the peptides with a higher scoring (best Conf-Peptide) were preferred, representing a higher confidence level; and (3) the secondary mass spectra of the parent ions were analyzed as follows: the complementarity of the N-terminal fragment ions with the C-terminal fragment ions. The extracted ion flow diagram-XIC diagram was drawn for the obtained peptides according to the principles, and the best peptides were screened according to the XIC diagram. The total ion flow diagram of the LC-MS/MS of the XSM peptide is shown in Figure 2.

As shown in Appendix A, a total of 10 best peptides of XSM were screened and identified, and these peptides were AEQPPLFDGT, GMVRELIVNVG, LCLELVNGVI, GGLPGFDPA, VTYPIIADPN, IMAVLAIVL, INPILLPK, ETYFIQ, TVWPGIQPN, and IAICNGVL. The scores of peptides, the molecular weights, retention times and peak areas are shown in Table 2. The 10 best peptides obtained from the XSM were used for simulated molecular docking with ACE.

### 2.4. Toxicity and Allergenicity Prediction of Peptides

The 10 peptides identified were evaluated for toxicity and allergenicity and the results are shown in Table 2. Toxicity is one of the fundamental factors that should be considered for various food and drug products. We evaluated the potential toxicity of 10 peptides using Toxinpred and the results showed that all peptides were nontoxic. The AllerTOP v.2 allergenicity assessment of the peptides showed that two peptides (LCLELVNGVI and TVWPGIQPN) showed potential allergenicity, while the other eight peptides were not potentially allergenic. Thus, the 10 peptides identified exhibited satisfactory or relatively satisfactory safety properties, and all of them are valuable for further studies.

### 2.5. Molecular Docking of the ACE Inhibitory Peptide

Enzyme and small molecule interactions were predicted by molecular docking. Based on a better understanding of the molecular mechanisms supporting ACE and inhibiting peptide interactions, the design or synthesis of novel ACE inhibitors may become easier [28]. Previous results have shown that ACE has three main active pockets (subsites), namely S1, S1′, and S2. Wu et al. [29] found that the ACE inhibitory peptides they obtained interacted with three residues in the S1 pocket, namely Glu384, Tyr523, and Ala354, with five residues in the S2 pocket, including Tyr520, His513, Gln281, Lys511, and His353, and only Glu162 in the S1′ pocket had significant interactions. These regions are responsible for the biological activity of ACE and are also the regions to which ACE inhibitors bind [30]. H-bonds are reported to be one of the most important noncovalent interactions for binding potential inhibitors of ACE [31]. We used the Autodock 1.5.7 software to perform molecular docking of the 10 identified peptides of XSM with known ACE active sites. As shown in Table 2, the docking energies of the best binding conformations of the 10 peptides and ACE were obtained based on the minimum binding energy after docking, in which the GGLPGFDPA, IMAVLAIVL, ETYFIVR, INPILLPK, and IAICNGVL docking energies were <0. We further analyzed these five peptides.

According to the docking results, the binding energy of ETYFIVR with the most stable binding to the ACE active site was −3.38 Kcal/mol. Figure 3A shows that the peptide ETYFIVR formed two coordination bonds with Zn^2+^ in the ACE active center, while 12 hydrogen bonds were established with ACE amino acid residues, and nine hydrogen bonds were established with the ACE active center. Among them, ETYFIVR formed hydrogen bonds with Ala354 and Tyr523 in the S1 pocket with lengths of 2.5 Å and 1.9 Å, respectively, and with His513, Gln281, His353, and Lys511 in the S2 pocket with hydrogen bond lengths of 2.1 Å, 2.5 Å, 2.9 Å, and 2.2 Å, respectively. Notably, ETYFIVR formed two hydrogen bonds with Glu162 in the S1′ pocket with hydrogen bond lengths of 2.87 Å and 3.20 Å. Glu162 formed two hydrogen bonds with GGLPGFDPA, indicating that Glu162 contributed the most to the formation of the stable conformation. Therefore, we suggest that Lys511 and Glu162 in ACE are the most important amino acid residues for the stable binding of GGLPGFDPA to ACE. The hydrogen bonds formed by the amino acid residues around the active center with the peptide GGLPGFDPA generate a tighter and more stable bond between the peptide and ACE.

The binding energy of GGLPGFDPA with its most stable binding to the active site of ACE was −0.53 Kcal/mol. As seen in Figure 3B, GGLPGFDPA established the presence of 11 hydrogen bonds with ACE residues, and five hydrogen bonds were bound to the active site of ACE. Among them, GGLPGFDPA formed hydrogen bonds with Gn281, His353, His520, and Lys511 in the S2 pocket. Specifically, GGLPGFDPA formed two hydrogen bonds with Lys511 with lengths of 2.1 Å and 2.4 Å and one hydrogen bond each with Gn281, His353, and Tyr520 with lengths of 1.7 Å, 2.1 Å, and 2.0 Å, respectively. Lys511 formed two hydrogen bonds with GGLPGFDPA, indicating that Lys511 contributed the most to the formation of the stable conformation. Therefore, we suggest that Lys511 in ACE is the most important amino acid residue for the stable binding of GGLPGFDPA to ACE. The hydrogen bonds formed by the amino acid residues around the active center with the peptide GGLPGFDPA result in a tighter and more stable bond between the peptide and ACE.

The binding energy of peptide IAICNGVL at the most stable binding with ACE active site is −5.42 Kcal/mol. Figure 3C shows that peptide IAICNGVL established the presence of 2 hydrogen bonds with ACE residues and no hydrogen bond with the ACE active site. Therefore, it is presumed that the peptide IAICNGVL does not have ACE inhibitory activity.

The binding energy of IMAVLAIVL at its most stable binding to the active site of ACE is −5.38 Kcal/mol. As seen in Figure 3D, IMAVLAIVL establishes the presence of eight hydrogen bonds with ACE residues and five hydrogen bonds with the active site of ACE. IMAVLAIVL forms hydrogen bonds with Ala354 and Glu384 in the S1 pocket with lengths of 1.8 Å, and 1.6 Å, respectively, and forms hydrogen bonds with Gln281, His353, and His513 in the S2 pocket, of which two hydrogen bonds with His353 with lengths of 2.3 Å and 2.7 Å and one hydrogen bond with each of Gln281 and His513 with lengths of 2.8 Å and 1.9 Å, respectively. His353 formed two hydrogen bonds with GGLPGFDPA, indicating that His353 contributed the most to the formation of the stable conformation. Therefore, we suggest that His353 in ACE is the most important amino acid residue for the stable binding of GGLPGFDPA to ACE. The hydrogen bonds formed by the amino acid residues around the active center with the peptide GGLPGFDPA result in a tighter and more stable bond between the peptide and ACE.

The binding energy for the most stable binding of INPILLPK to the active site of ACE was −1.73 Kcal/mol. As seen in Figure 3E, INPILLPK established the presence of six hydrogen bonds with ACE residues, and three hydrogen bonds were bound to the active site of ACE. Among them, INPILLPK formed hydrogen bonds with Tyr523 and Glu384 in the S1 pocket with bond lengths of 2.2 Å and 2.5 Å, respectively, and with His353 in the S2 pocket with a hydrogen bond length of 2.4 Å. It can be seen that the hydrogen bond formed by INPILLPK with His353 is the shortest, indicating that His353 contributes most to the formation of the stable conformational phase. Therefore, we suggest that His353 in ACE is the most important amino acid residue for the stable binding of INPILLPK to ACE. The hydrogen bonds formed by the amino acid residues around the active center and the peptide INPILLPK make the peptide bind to ACE more tightly and stably.

Based on the results, GGLPGFDPA, IMAVLAIVL, ETYFIVR, and INPILLPK can all bind to the active site of ACE, so we selected these four peptides to synthesize and evaluate their inhibitory effects on ACE. Notably, ETYFIVR binds the most hydrogen bonds to the ACE active site and forms ligand bonds with Zn^2+^. Zn^2+^ at the ACE active site usually plays an important role in ACE activity, and the binding of Zn^2+^ to peptide receptors is a key factor in ACE inhibitory activity [30]. Chen et al. [15] investigated the mechanism of ACE inhibition by the peptides EACF and CDF in rabbit protein and found that EACF formed eight hydrogen bonds with the S1 and S2 active pockets of ACE and ligand bonds with Zn^2+^, and CDF formed four hydrogen bonds with the S1 active pocket of ACE; in addition, the researchers found that EACF had a low IC_50_. Therefore, it can be speculated that ETYFIVR has the highest ACE inhibition rate among the four peptides.

### 2.6. In Vitro Inhibition Rate Activity of the Synthetic Peptides

The peptides were synthesized by solid-phase synthesis and purified by HPLC, and the structures were determined by LC-MS/MS. GGLPGFDPA, ETYFIVR, and INPILLPK were successfully synthesized, while IMAVLAIVL could not be synthesized due to the presence of too many hydrophobic amino acid residues.

Figure 4A demonstrates the in vitro inhibitory activity of GGLPGFDPA, ETYFIVR, and INPILLPK. The results showed that ETYFIVR had the highest ACE inhibitory activity of 67.02 ± 1.25%, while GGLPGFDPA was 24.89 ± 0.83% and INPILLPK was 4.19 ± 0.62% at a concentration of 0.1 mg/mL. The high inhibitory activity of ETYFIVR corroborated the results of molecular docking, in which the seven hydrogen bonds formed with the ACE active pocket and the ability to form ligands with Zn^2+^ resulted in a high activity of ACE inhibition. A 24.89 ± 0.83% ACE inhibitory activity was also obtained with GGLPGFDPA, in which five hydrogen bonds formed with the ACE active pocket. However, the IMAVLAIVL inhibitory activity was only 4.19 ± 0.62%, which may be due to its formation of only three hydrogen bonds with the ACE active pocket. In summary, the ACE inhibition rates of the synthetic peptides were in high agreement with the molecular docking results, indicating that the method of screening ACE inhibitory peptides in this study is effective and reliable and that ETYFIVR and GGLPGFDPA are two bioactive peptides with the potential to inhibit ACE activity.

### 2.7. Synthetic Peptide Anti-Gastrointestinal Enzyme Stability

For some peptides, their ACE inhibition rate activity was reduced or even completely lost by digestive enzyme treatment [32]. Therefore, it is of interest to study the changes in peptide activity after gastrointestinal digestion. The GGLPGFDPA and ETYFIVR peptides with high ACE inhibitory activities were selected for gastrointestinal simulation experiments.

As shown in Figure 4B, the inhibitory activity of ETYFIVR decreased from 67.02 ± 1.25% to 61.43 ± 0.67% after pepsin digestion treatment, and the inhibitory activity did not decrease significantly, indicating that ETYFIVR could maintain high stability under pepsin digestion conditions. However, after pepsin treatment, the inhibition of GGLPGFDPA increased from 24.89 ± 0.83% to 41.48 ± 1.26%. This may be because pepsin can digest GGLPGFDPA, releasing smaller peptides and thus enhancing ACE inhibitory activity [33].

Figure 4C shows the ACE inhibitory activity of GGLPGFDPA and ETYFIVR after pepsin and trypsin treatment. The inhibitory activity of ETYFIVR was reduced from 61.43 ± 0.67% to 30.86 ± 1.98% after further trypsin treatment. This result is similar to that of Chen et al. [34]. The results of Chen’s study indicated that peptides tend to be hydrolyzed by intestinal digestion rather than inactivated in the stomach. Notably, the inhibition of GGLPGFDPA increased from 41.48 ± 1.26% to 49.35 ± 4.62% after further trypsinization, possibly because trypsin can continue to hydrolyze the small molecule peptides that are produced by pepsin hydrolysis; thus, more ACE active peptides are released. This is similar in nature to the TQPKTNAIPY peptide fragment that was obtained by Gómez-Ruiz et al. [35].

In conclusion, ETYFIVR at 0.1 mg/mL exhibited an ACE inhibition activity of 30.86 ± 1.98% despite instability under gastrointestinal digestive enzyme treatment (67.02 ± 1.25% before gastrointestinal digestion); GGLPGFDPA at 0.1 mg/mL exhibited a significant ACE inhibition activity after gastrointestinal digestive enzyme treatment, eventually exhibiting 49.35 ± 4.62% ACE inhibition (24.89 ± 0.83% before gastrointestinal digestion). Xu et al. [36] screened six peptides from soybean isolate protein with ACE inhibition activities of 66.26–93.53% at 1 mg/mL, and after gastrointestinal digestion, six peptides ace inhibition activities of 7.65–90.16%; He et al. [37] identified two ACE inhibitory peptides from yellow wine lees with ace inhibition activities of 89.27% and 97.12% at 1 mg/mL, and after gastrointestinal digestion, the two peptides ace inhibition decreased to 87.67% and 89.05%. However, ETYFIVR and GGLPGFDPA screened in this study showed good ACE inhibition at a concentration of 0.1 mg/mL. Therefore, ETYFIVR and GGLPGFDPA are considered as two bioactive peptides with anti-hypertensive potential.

## 3. Materials and Methods

### 3.1. Materials and Reagents

*X. sorbifolium* seeds were collected from the Weng Niu Te economic forest farm, Chifeng city, Inner Mongolia Autonomous Region. Hippuric acid, hippuryl-histidyl-leucine (HHL) and angiotensin-I-converting (ACE) enzymes were provided by Sigma Chemical Co. (St. Louis, MO, USA). Alkaline protease (5 × 10^4^ U/g) was purchased from Aobox Biotechnology Co. (Beijing, China). Other reagents used in the experiments were of analytical grade and purchased from Beijing Chemical Plant Co. (Beijing, China). All materials were stored for a maximum of 2 weeks.

### 3.2. Screening of Hydrolytic Enzymes

Computerized virtual hydrolysis of known proteins was performed using BIOPEP (http://www.uwm.edu.pl/biochemia/index.php/pl/biopep#opennewwindow accessed on 15 December 2021) to screen for hydrolytic proteases in the XSM proteins. Albumin is the main protein in XSM, and the amino acid composition of XSM protein is most similar to that of soybean [10]. Therefore, soybean albumin was used as a bioinformatic template for the XSM proteins. The following known albumin sequences were searched in the UniProt Knowledgebase (https://www.UniProt.org/ accessed on 28 December 2021): 2SS_SOYBN (UniProt ID: P19594); ALB1_SOYBN (UniProt ID: Q39837); ALB1_PSOTE (UniProt ID: P15465), and virtual hydrolysis was performed online using the built-in software in the database. We selected alkaline protease (EC 3.4.21.62), pepsin (EC 3.4.23.1), trypsin (EC 3.4.21.4), V-8 protease (EC 3.4.21.19) and plasmin (EC 3.4.21.7), which are contained in the UniProt database and commonly used in the laboratory, to perform virtual hydrolysis of the three albumin sequences. The built-in software automatically calculates the corresponding degree of hydrolysis and the frequency of ACE inhibitory activity (AE) release for each protease.

### 3.3. Isolation of Crude Protein Extract

*X. sorbifolia* seeds were ground into powder using a grinder (Model FW100, Taisite Co., Tianjin, China) and passed through a screener of mesh number 20. The *X. sorbifolia* powder was blended with distilled water (1:35 *w*/*v*), and the pH was adjusted to 11 with 1 mol/L NaOH. Then, this solution was placed in the ultrasonic instrument and extracted at 40 °C for 40 min, and the power was 100 W. After centrifugation was performed at 5000 r/min for 20 min, the pH of the supernatant was adjusted to 4.5 with 0.5 mol/L HCl to precipitate the protein. The precipitated proteins were collected and dissolved in distilled water, and then the pH was adjusted to 7.0. After freeze-drying, the *X. sorbifolia* proteins served as the sample for subsequent enzymatic hydrolysis.

### 3.4. Process Optimization for the Proteolytic Hydrolysis of XSM

#### 3.4.1. Hydrolysis Determination

The method was based on formaldehyde titration [38]. The hydrolysate (8 mL) of XSM protein hydrolyzed by alkaline protease was mixed with 60 mL deionized water and 10 mL formaldehyde solution, and the pH was adjusted to 9.2 with 0.05 mol/L NaOH. The controlled experiment was conducted as follows: the above steps were repeated with 8 mL of the same concentration of unhydrolized protein solution, and the consumed volume was recorded.

The DH of the protein was calculated using the following equation:DH (%) = M (V_1_ − V_2_) × 1000 × (1/(CV)) × (1/H_tot_) × 100(1)
where M = concentration of NaOH standard solution (mol/L), V_1_ = volume of 0.05 mol/L NaOH standard solution of hydrolysate (mL), V_2_ = volume of 0.05 mol/L NaOH standard solution of unhydrolysate (mL), C = concentration of enzymolysis solution and protein solution (g/L), V = volume of enzymolysis solution and hydrolysate (mL), and H_tot_ = number of peptide bonds in substrate proteins.

#### 3.4.2. ACE Inhibitory Activity Assay

The HPLC apparatus was as follows: a Shimadzu LC-10ATvp system HPLC (Shimadzu, Kyoto, Japan) equipped with an LC-10AT pump (Shimadzu), a Sil-10A autosampler (Shimadzu), an SPD-M10A DAD, and a Shimadzu Class-VP 6.12 chromatographic data processor was used. The analytical column was a C18 column (250 mm × 4.6 mm.d × 5 μm, Shimadzu). The mobile phase was a mixture of methanol and water (50/50, volume ratio), and the flow rate was 1.0 mL/min. The detection wavelength was 228 nm, and the injection volume was 20 μL [39].

ACE inhibitory activity was measured by high-performance liquid chromatography (HPLC) to determine the potential antihypertensive activity. Samples were prepared using 100 mmol/L sodium borate buffers (pH 8.3). The samples contained 30 μL of 5 mmol/L Hippuryl-His-Leu-OH (HHL) and 20 μL of the enzymatic hydrolysate as a substrate. After incubation for 3 min at 37 °C, 20 μL of ACE powder (0.1 units) was added to this tube. Finally, the mixture was incubated for 30 min at 37 °C, and the reaction was terminated by adding 60 μL of 1 mol/L HCl. Sodium borate buffers (100 mmol/L) were included as a blank control.

The ACE inhibitory activity was calculated using the following equation:ACE inhibitory activity% = (A − B)/A × 100%(2)
where A = peak area of hippuric acid in the blank group, and B = peak area of the reaction solution at the retention time of hippuric acid.

#### 3.4.3. Single-Factor Experiments

A certain mass of protein powder was weighed to prepare a protein solution with a substrate concentration of 6%, and the preparation procedure was optimized using the following parameters: hydrolysis temperature (25 °C, 30 °C, 35 °C, 40 °C, 45 °C); pH (8.0, 8.5, 9.0, 9.5, 10.0); amount of protease added (1%, 2%, 4%, 6%, 8%) (enzyme mass / protein powder mass); and hydrolysis time (0 min, 10 min, 20 min, 30 min, 40 min, 50 min). The enzyme solution was inactivated at 100 °C for 10 min and cooled to room temperature. HCl (1 mol/L) was used to adjust the pH value to an isoelectric point of 4.6 for the XSM protein, and the peptide was centrifuged at 5000 r/min for 15 min. The supernatant was aspirated, adjusted to pH 7, and freeze-dried to obtain the peptide.

#### 3.4.4. Orthogonal Experiments

An orthogonal L9(3^4^) test design was used for the optimization of XSM protein hydrolysates. Orthogonal experiments were performed using the following parameters: hydrolysis temperature (35 °C, 40 °C, 45 °C); pH (8.5, 9.0, 9.5); amount of protease added (3%, 4%, 5%); and hydrolysis time (15 min, 20 min, 25 min).

### 3.5. Identification and Screening of the Structure of Potential Peptides from XSM by LC-MS/MS

#### 3.5.1. Sample Preparation

The lyophilized peptide samples were dissolved in a PBS phosphate buffer solution (0.1 mol/L, pH 8.0) and centrifuged several times until the supernatant was clear and transparent, and the sample peptides were diluted with 0.1% formic acid solution (containing 2% acetonitrile) before analyzing the liquid quality.

#### 3.5.2. Chromatographic Analysis

The chromatographic conditions were as follows: precolumn: ChromXP C18, 3 μm, 120 A; analytical column: ChromXP C18, 150 × 0.30 mm, 3 mm, 120 Å pore size. The loading pump delivered 0.1% formic acid aqueous solution at 10 μL/min, and the micropump produced a flow rate of 5 μL/min. The elution conditions were as follows: mobile phase A: 98% water, 2% acetonitrile, 0.1% formic acid; mobile phase B: 98% acetonitrile, 2% water, 0.1% formic acid; injection volume: 15 µL; and column temperature: 30 °C. The starting mobile phase conditions were A = 97% and B = 3%. At 5.0 min A = 95% and B = 5%, and at 30.0 min A = 85% and B = 15%. At 50.0 min A = 75% and B = 25% and at 55 min A = 65% and B = 35%. At 70.0 min A = 55% and B = 45%, and at 75.0 min A = 50% and B = 50% which was held for 7 min. At 82.1 min the initial conditions were reinstated and column reequilibration was achieved after an additional 1 min (total run time = 83.1 min).

The mass spectrometry conditions were as follows: ionspray voltage floating (ISVF) 5500 V, curtain gas (CUR) 25, collision energy (CE) 10 and ion source gas 1 (GS1) 25. The switching criteria were set to ions greater than the mass-to-charge ratio (*m*/*z*) 300 and were smaller than *m*/*z* 1300 with a charge state of 2–5, mass tolerance of 50 ppm and an abundance threshold of more than 100 counts (cps). Former target ions were excluded for 15 s.

#### 3.5.3. Data Acquisition

The instrument data acquisition mode was set in the mass range of 300–1300 m/z to obtain high resolution (30,000) TOF-MS scans with up to 40 (first 40) high sensitivity MS/MS scans per cycle for the most abundant peptide ions. The dynamic exclusion duration was set to 12 s to account for differences in peak widths and peak matches. Each measurement (TOF-MS) scan lasted 0.25 s for a total cycle time of 2.3 s. Ions were fragmented in the collision cell using rolling collision energy with CES set to 5. The collected peptide ion fragmentation spectra were stored in a wiff format (SCIEX).

### 3.6. Toxicity and Allergenicity Prediction of Peptides

Some properties of the peptide are crucial to determine whether this peptide is of value for further research, such as toxicity and sensitization. Therefore, to further evaluate the peptides obtained from the identification, we need to obtain the toxicity and sensitization properties of the peptides. The toxicity of the peptides was analyzed by ToxinPred (http://crdd.osdd.net/raghava/toxinpred/ accessed on 8 January 2022). AllerTOP v.2 (http://www.ddg-pharmfac.net/AllerTOP/index.html accessed on 8 January 2022) was used to analyze the allergenicity of the peptide.

### 3.7. Molecular Docking of ACE Binding Sites

The receptor used in this study was the 3D crystal structure of the human ACE-lisinopril complex (1O8A.pdb), which was downloaded from the RCSB PDB protein database (http://www.rcsb.org/pdb/home/home.do, accessed on 8 January 2022). The peptides structures were drawn using the Chem3D 20.0 package and converted to 3D structures in the Chem3D 20.0 software. Molecular docking of the ACE inhibitory peptides and ACE was performed using the AutoDock software. AutoDock combines an empirical free energy force field with a Lamarckian Genetic Algorithm, providing fast prediction of bound conformations with predicted free energies of association. Before docking analysis, water molecules and the inhibitor lisinopril were removed using PyMOL (2.3.0), while the cofactors zinc and chloride atoms were retained in the active site of the crystal structure of ACE. Docking simulations were performed using the Autodock Tools (Version 1.5.7). The docking runs were carried out as follows: coordinates x: 41.586, y: 37.383, and z: 43.445; grid box size of 90 Å × 90 Å × 90 Å. The Lamarck genetics algorithm was used to dock each peptide small molecule structure by flexible docking, and the ligand with the lowest binding energy in the protein binding bag was selected as the best docking posture. The best-scoring pose judging by the Vina docking score was selected and visually analyzed in the PyMoL 1.7.6 software.

### 3.8. Peptide Synthesis

To investigate the in vitro ACE inhibitory activity of the peptides screened by molecular docking, we chose the solid-phase peptide synthesis method (Divbio Pharmaceutical Co., LTD, Shenzhen, China) to synthesize the peptides. The purity and amino acid sequence of the synthesized peptides were determined by HPLC and LC-MS, respectively. The purity of the peptides used in this study was >99%.

### 3.9. Stability of ACE Inhibitory Peptides during In Vitro Simulated Digestion

The simulated digestion method was performed according to the method of Xu et al. [36] with some modifications. Briefly, the synthesized peptides were diluted with 10% acetonitrile solution to a final concentration of 0.1 mg/mL and were adjusted to pH 2.0 with HCl (0.1 M). The resulting peptide solution was added to pepsin (final concentration: 2%, *w*/*w*), and the mixture was then placed in a water bath (37 °C) for 60 min. After the gastric mock digestion was complete, the reaction was interrupted by adjusting the pH to 8.0 with NaOH (1 M) to obtain mixture A. A portion of mixture A was aspirated, pancreatin (final concentration: 2%, *w*/*w*) was added, and then the mixture was placed in a water bath (37 °C) for 60 min. After the intestinal digestion was complete, the pH of the mixture was adjusted to 7.0. Then, the mixture was placed in a boiling water bath to inactivate the enzyme for 10 min, cooled to room temperature and frozen at −20 °C. This part of the solution was recorded as mixture B.

The ACE inhibition activities of mixture A and mixture B were measured separately to determine the stability of the synthetic peptide to the gastrointestinal enzyme.

### 3.10. Statistical Analysis

All experiments were performed in triplicate, and the results are reported as the mean ± standard deviation (SD). Statistical differences between the control and experimental groups were determined by one-way ANOVA using the SPSS Statistics for Windows version 13.0 (SPSS, Chicago, IL, USA); the results were considered statistically significant at a p value less than 0.05.

## 4. Conclusions

In this study, bioactive peptides with high ACE inhibition rates were successfully identified from XSM using a computer-assisted experimental screening method. First, to obtain more hydrolysis mixtures that contained active peptides with ACE inhibition rates, BIOPEP-UWM, UniProt and orthogonal experiments were utilized to optimize the hydrolysis conditions of the XSM proteins. The optimal conditions were a 5% enzyme dosage, 45 °C, 15 min, and pH 9.5. Subsequently, the 10 best peptides in the hydrolysate were screened using LC-MS/MS and these 10 best peptides exhibited relatively satisfactory safety properties. Molecular docking results showed that GGLPGFDPA, IMAVLAIVL, ETYFIVR, and INPILLPK were the four peptides with ACE inhibitory potential, and ETYFIVR had the highest ACE inhibitory activity. Then, ETYFIVR, GGLPGFDPA, and INPILLPK were successfully synthesized by solid-phase synthesis, and their ACE inhibition rates were determined to be 67.02 ± 1.25%, 24.89 ± 0.83%, and 4.19 ± 0.62%, respectively. This result is in agreement with the molecular docking results. Finally, it was shown through gastrointestinal digestion simulation experiments that ETYFIVR inhibition decreased from 67.02 ± 1.25% to 30.86 ± 1.98% and GGLPGFDPA inhibition increased from 24.89 ± 0.83% to 49.35 ± 4.62% after pepsin and trypsin treatment. This indicates that ETYFIVR and GGLPGFDPA have a certain stability under the treatment of gastrointestinal digestive enzymes. However, the regulation of the renin-angiotensin system involves several peptides and substrates, so testing the balance of ETYFIVR and GGLPGFDPA in all systems will be our future research direction.

In this study, it was shown that XSM proteins could be suitable materials for the preparation of ACE inhibitory peptides, and ETYFIVR and GGLPGFDPA are two novel potential ACE inhibitory peptides that are present in XSM. Computer-assisted experimental screening is an efficient, accurate and promising method for the discovery of bioactive peptides.

## Figures and Tables

**Figure 1 molecules-27-08792-f001:**
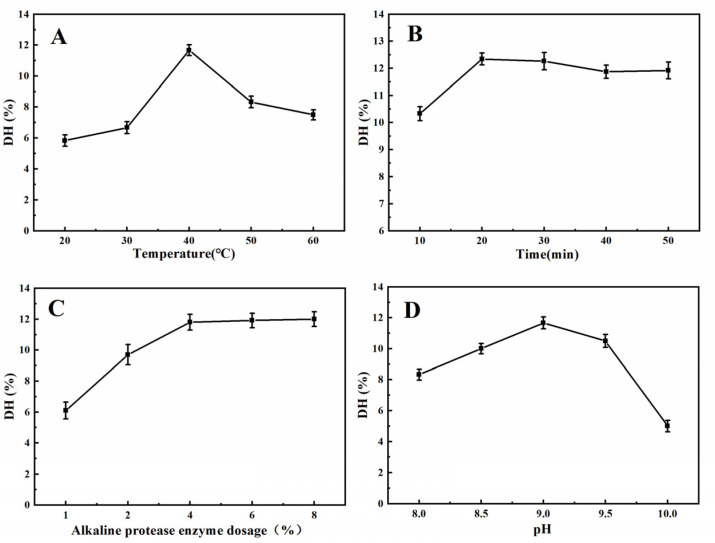
Influence of the hydrolysis parameters on the degree of hydrolysis of the XSM proteins. (**A**) Temperature. (**B**) Time. (**C**) Alkaline protease enzyme dosage. (**D**) pH.

**Figure 2 molecules-27-08792-f002:**
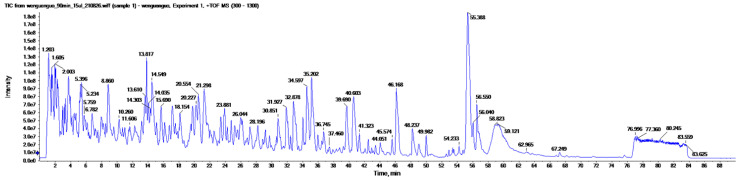
Total ion flow diagram of peptides from XSM.

**Figure 3 molecules-27-08792-f003:**
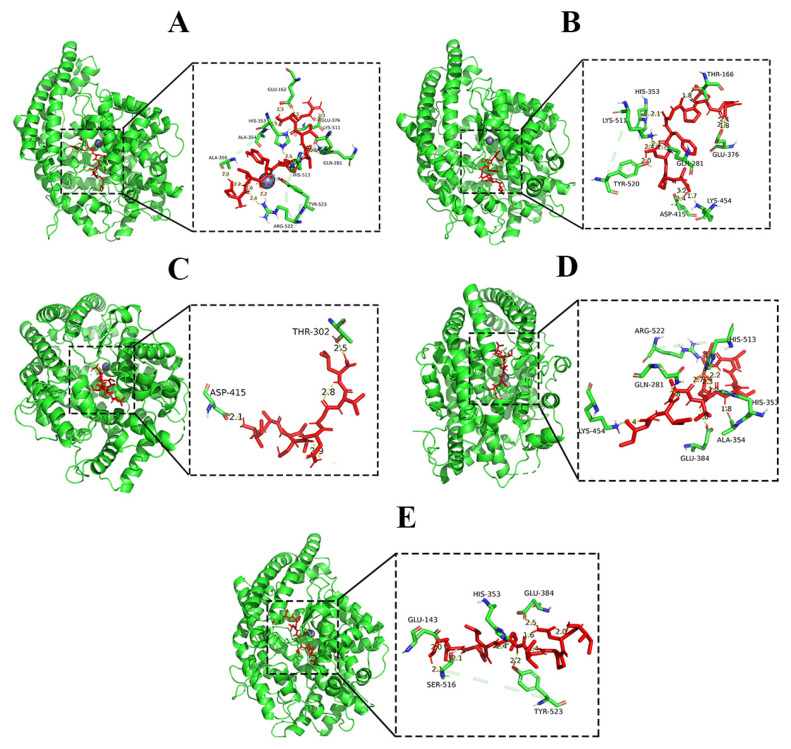
Molecular docking results for ETYFIVR, GGLPGFDPA, IAICNGVL, IMAVLAIVL and INPILLPK binding with ACE molecule. (**A**) ETYFIVR. (**B**) GGLPGFDPA. (**C**) IAICNGVL. (**D**) IMAVLAIVL. (**E**) INPILLPK.

**Figure 4 molecules-27-08792-f004:**
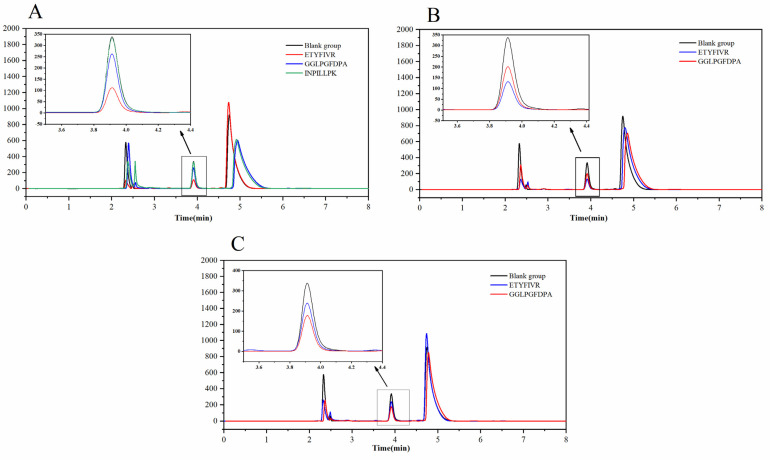
Chromatogram of ACE inhibitory activity of synthetic GGLPGFDPA, ETYFIVR and INPILLPK in vitro (**A**). ETYFIVR and GGLPGFDPA after pepsin treatment (**B**). ETYFIVR and GGLPGFDPA after pepsin and trypsin treatment (**C**).

**Table 1 molecules-27-08792-t001:** Simulated results of hydrolysis rate and ability to release ACE inhibitory peptides of soy albumin treated with five enzymes.

Enzyme Protein	2SS_SOYBN ^1^		ALB1_SOYBN ^2^		ALB1_PSOTE ^3^	
	DH ^4^ (%)	AE ^5^	DH (%)	AE	DH (%)	AE
Alkaline protease	25.58	0.0227	16.19	0.0115	24.35	0.0155
Pepsin	11.54	-	9.25	-	8.29	0.0052
V-8 protease	3.88	-	9.83	0.0057	4.17	-
Trypsin	7.52	-	13.87	0.0015	11.40	0.0052
Plasmin	6.98	-	13.95	0.0116	11.40	0.0052

^1^ 2SS_SOYBN, ^2^ ALB1_SOYBN, ^3^ ALB1_PSOTE: the albumin sequences were searched in the UniProt Knowledgebase (https://www.UniProt.org/ accessed on 28 December 2021): UniProt ID: P19594 (2SS_SOYBN); UniProt ID: Q39837 (ALB1_SOYBN); UniProt ID: P15465 (ALB1_PSOTE).; ^4^ DH: hydrolysis degree of protein.; ^5^ AE: the frequency of ACE inhibitory activity release.

**Table 2 molecules-27-08792-t002:** LC-MS/MS identification results, safety characteristics and docking energies with ACE for 10 peptides in XSM proteins.

Sequence	Scores	Theor m/z (Da)	Retention Times	Peak Areas	Toxicity	Allergenicity	Docking Energies (Kcal/mol)
AEQPPLFDGT	96.3	537.7587	30.951	3611.79	Non-Toxin	Non-allergen	18.86
GMVRELIVNVG	57.4	396.2253	35.259	2180.64	Non-Toxin	Non-allergen	22.38
LCLELVNGVI	52.4	358.2072	32.474	19,001.02	Non-Toxin	Allergen	6.79
GGLPGFDPA	25.8	415.7058	33.917	4160.98	Non-Toxin	Non-allergen	−0.53
VTYPIIADPN	29.9	551.7926	33.584	23,527.62	Non-Toxin	Non-allergen	12.27
IMAVLAIVL	50.9	471.8065	38.362	2554.76	Non-Toxin	Non-allergen	−5.38
INPILLPK	31.6	454.3024	31.137	2336.7	Non-Toxin	Non-allergen	−1.73
ETYFIVR	32.1	464.2504	31.709	3716.81	Non-Toxin	Non-allergen	−3.38
TVWPGIQPN	27.1	506.2665	31.287	9670.05	Non-Toxin	Allergen	4.11
IAICNGVL	19.6	401.7282	10.161	9542.85	Non-Toxin	Non-allergen	−5.42

## Data Availability

Not applicable.

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
