# Peer review of "Screening and Mechanism of Novel Angiotensin-I-Converting Enzyme Inhibitory Peptides in X. sorbifolia Seed Meal: A Computer-Assisted Experimental Study Method"

_molecules, 2022, doi:10.3390/molecules27248792_

Round 1

Reviewer 1 Report

In general the results are well presented but the English could be improved - particularly the methods section.  There is a difference between 'the' and 'a', which the authors do not appreciate.

This paper suffers from some sweeping statements that are not supported by the data.  For example (line 264) 4 peptides were found to bind to the active site of ACE and therefore have ACE inhibitory potential.  The binding is weak the peptides were found not to all have ACE inhibition.

The authors use the term  inhibitory rate and inhibitory activity interchangeably, when in fact no time based study was done.  

The experimental details of the molecular modelling are very scant.  Only the software is given.  How was the docking done?  What about the flexibility of the peptides.  Were multiple poses collected and grouped?

The binding of the peptides is weak.  How do they compare with other binding studies?

The inhibitory activity of GGLPGFDPA increased after direction.  Was no attempt made to identify the hydrolysis product - presumably the smaller peptides would be better as the authors postulate that they have enhanced activity.

Reviewer 2 Report

The authors present the advantages found in the use of computer-assisted experimental approachs, particularly in the research of ACE inhibitory peptides. They recall some other studies where isolated peptides of different food or organisms have been found as ACE inhibitors, and also comment on similar studies that used computer-assistance (Ali et al, 2020; Zheng et al, 2019; Liu et al, 2021).

The sentence of line 92 can include the five enzymes that were tested and the three albumins selected, instead of “Table 1” and “several albumins selected”. Some of the information is included in Materials and Methods, but the manuscript version should be prepared from the begining according to the instructions for authors (it seems that the manuscript was prepared with another format, where materials and methods are listed previous to the results section). It is not clear in the results section if there were other albumins included in the research, and only the results for three of them were selected (as described in Materials and Methods). The methodology for selection of the five enzymes could also be described.

Line 98 includes the title of Table 1, which could be placed in line 99; this reviewer recomends that table 1 includes the abbreviation meanings.

For the description of hydrolysis process, the authors need to clarify which enzyme was used (Fig.1), if it was the selected alkaline protease as described, it should be clear. Materials and Methods section is until line 331, and difficult to search for this information.

Figure 1C is not about enzyme dosage, authors need to revise the results section and verify every data involved. Statement in line 119 is not precise; authors need to explain the calculations, so that results are in concordance to DH (if “products” obtained from “substrate” –reactants- were measured directly, this aseveration would not be correct). Description of Figure 1D is also incorrect, for pH influence was showed in Figure 1C.

It was not discussed that enzyme selection could use non-mammalian enzymes, which have other optimal temperature; the influence of pH was neither discussed in the preselection of enzymes, considering that pepsin was one of the pre-selected enzymes, and optimal pH for its activity is not comparable to alkaline protease.

For docking analysis, this reviewer suggest to clarify if interactions were searched only for orthosteric inhibition, the three active pockets, or the wide search for interactions was registered, considering that allosteric inhibition can be found and docking energies reported for several peptides are not good enough to be considered.

Some details of format could be improved, examples: capital letters in line 336, citation format in line 344, “consumed volume was recorded” in line 370, “need” line 445, “comceptualization”.

In supplementary material, tableS1 needs to correct format, so “D” is on line with A,B and C.

The manuscript is somehow similar to the following non-cite article: Wei Y, Liu Y, Li Y, Wang X, Zheng Y, Xu J, Sang S, Liu Y. A Novel Antihypertensive Pentapeptide Identified in Quinoa Bran Globulin Hydrolysates: Purification, In Silico Characterization, Molecular Docking with ACE and Stability against Different Food-Processing Conditions. Nutrients. 2022 Jun 10;14(12):2420. doi: 10.3390/nu14122420. PMID: 35745149; PMCID: PMC9227351.

This reviewer considers that this is an original publication, although novelty of the findings is compromised, exception made for the seed meal; authors can improve the article discussing (or comparing) the findings for these peptides with the existing ACE inhibitors, considering that the regulation of the renin-angiotensin system involves several peptidases and substrates (namely hormones), and the effectiveness of new inhibitors shall be tested for the homeostasis of all the system.
